# Healthcare Provider-Based Contraceptive Coercion: Understanding U.S. Patient Experiences and Describing Implications for Measurement

**DOI:** 10.3390/ijerph21060750

**Published:** 2024-06-08

**Authors:** Laura E. T. Swan, Lindsay M. Cannon

**Affiliations:** 1Department of Population Health Sciences, University of Wisconsin-Madison, Madison 53706, WI, USA; 2Department of Sociology, Center for Demography and Ecology, University of Wisconsin-Madison, Madison 53706, WI, USA; lmcannon@wisc.edu

**Keywords:** contraceptive coercion, provider bias, discrimination, family planning, quality of care, patient-centered care

## Abstract

Despite growing concerns over coercion in contraceptive care, few studies have described its frequency and manifestations. Further, there is no established quantitative method of measuring this construct. We begin to fill this gap by detailing nuance in contraceptive coercion experiences and testing a novel measure: the Coercion in Contraceptive Care Checklist. In early 2023, we surveyed reproductive-aged people in the United States who were assigned female at birth about their contraceptive care. We describe the frequency of contraceptive coercion in our sample (*N* = 1197) and use open-ended descriptions to demonstrate nuances in these experiences. Finally, we debut our checklist and present psychometric testing results. Among people who had ever talked to a healthcare provider about contraception, over one in six participants (18.46%) reported experiencing coercion during their last contraceptive counseling, and over one in three (42.27%) reported it at some point in their lifetime. Being made to use or keep using birth control pills was the most common form of coercion reported by patients (14.62% lifetime frequency). Factor analysis supported the two-factor dimensionality of the Coercion in Contraceptive Care Checklist. Inter-item correlations were statistically significant (*p* < 0.001), providing evidence of reliability. The checklist was also related to measures of quality in family planning care (downward coercion: *t*[1194] = 7.54, *p* < 0.001; upward coercion: *t*[1194] = 14.76, *p* < 0.001) and discrimination in healthcare (downward coercion: *t*[1160] = −14.77, *p* < 0.001; upward coercion: *t*[1160] = −18.27, *p* < 0.001), providing evidence of construct validity. Findings provide critical information about the frequency and manifestations of contraceptive coercion. Psychometric tests reveal evidence of the Coercion in Contraceptive Care Checklist’s validity, reliability, and dimensionality while also suggesting avenues for future testing and refinement.

## 1. Introduction

Across U.S. history, social norms, governmental policies, and public health programs have promoted the reproduction of some groups, such as affluent white women, and restricted that of others, such as poor women and women of color. One overt example of racialized and classed stratified reproduction is the forceful and coercive sterilization of groups, including (but not limited to) poor white women, women with disabilities, and women of color, which was commonplace during the U.S. eugenics movement [1].

While sterilization abuse has become less common in recent years, public health policies and programs continue to promote stratified reproduction, although less explicitly. For example, economic and racial divides have been documented in “pre-pregnancy” care models that target poor women and women of color for reproductive life planning interventions while assuming that wealthy white women “embody reproductive responsibility” [2]. In another example, in the early 21st century, as initiatives such as the Affordable Care Act worked to eliminate barriers to contraceptive access [3], a new model of contraceptive counseling emerged—one that prioritizes contraceptive effectiveness and promotes long-acting reversible contraceptives (LARCs), including intrauterine devices (IUDs) and implants, above and before other contraceptive methods [4]. These models, known as tiered effectiveness counseling and LARC-first counseling, soon became recommended standards for contraceptive counseling [4].

Yet, scholars and advocates have criticized LARC-first models, critiquing the prioritization of providers’ concerns and those of medical governing bodies over patients’ needs and preferences [4,5]. LARC-first models can undermine patients’ priorities by placing primacy on effectiveness and ignoring other factors that are important in contraceptive decision-making, such as affordability and information about possible side effects [6].

Qualitative research has documented that patients can perceive providers’ overzealousness about LARCs as coercive [7,8,9]. Patient accounts have documented a range of contraceptive coercion practices, from more subtle examples of biased or directive counseling to more overt coercion such as outright refusal to remove a LARC method [10,11]. Patients also perceive the promotion of LARCs as more common among women of color, poor women, and women with less education [10].

Although most of the existing literature focuses on providers’ tendency to promote LARC methods, contraceptive coercion may encompass a variety of provider practices, including explicit or implied promotion or refusal of a range of contraceptive methods. Scholars have conceptualized *downward* coercion as pressure from a healthcare provider *not to use contraception*. For example, discouraging or refusing a wanted permanent contraception procedure, as documented in several studies [9,12,13,14], are examples of downward coercion. Conversely, *upward* coercion involves pressure from a healthcare provider *to use contraception* [11,15]. Forcible sterilization, as well as the promotion of or refusal to remove LARC methods, are examples of upward coercion.

Despite growing concerns over coercion and bias in contraceptive care, very few quantitative studies have described the frequency and manifestations of healthcare provider-based contraceptive coercion. Further, there is no established method of measuring contraceptive coercion. One of the few existing quantitative studies of contraceptive coercion focused specifically on the Appalachian region of the United States, finding that 37% of Appalachian women of reproductive age had ever experienced contraceptive coercion [15]. This study found that experiences of contraceptive coercion may undermine patients’ reproductive autonomy through a decreased likelihood of using their preferred contraceptive method(s) [15]. However, given the regional focus of this previous study, these findings may not reflect the experiences of patients across the country.

Given the recent attention to contraceptive coercion and calls for patient-centered care, it is critical to build an understanding of how contraceptive coercion manifests, how it can be rigorously measured, and its possible impacts. In this study, we begin to fill this gap in the first exploration of patients’ perceptions of contraceptive coercion in a national sample. Building on scholars’ theoretical descriptions of contraceptive coercion [11,15] and the emerging literature on the topic, we detail nuances in patients’ experiences of coercion in contraceptive care, debut and test a novel measure of contraceptive coercion, and make recommendations for future measurement of the construct.

## 2. Materials and Methods

In the spring of 2023, we used Prolific, an online platform with a national panel of thousands of vetted participants, to survey reproductive-aged people in the United States who were assigned female at birth. Existing evidence has shown the Prolific platform, which includes a variety of validity checks (e.g., bot detection, attention checks, etc.), to be a reliable method of data collection that can aid in quick and cost-effective study recruitment [16]. We used Prolific’s built-in prescreening to oversample racial/ethnic and gender and sexual minorities to ensure a diverse sample that included groups that are often underrepresented in survey research [17,18] and also theorized as more likely to experience contraceptive coercion [9,19].

We collected online survey responses until we reached our target sample size of 1500 participants. After their participation, we compensated those who completed the survey with USD 4 (a rate of USD 19 per hour based on the average response time). From the 1500 responses, we excluded 100 participants from the analytic sample because their responses to our validity check items indicated that they were not eligible for study participation (*n* = 7 were outside of the targeted age range of 18–49 years old; *n* = 93 were not assigned female at birth). We also excluded two participants with missing data on our key variables. Finally, our analytic sample was restricted to participants who reported ever talking to a healthcare provider about birth control (*N* = 1197); these were the only participants asked about contraceptive coercion, since those who have never experienced contraceptive counseling would not be at risk of experiencing contraceptive coercion. This research was approved by the Institutional Review Board at the University of Wisconsin-Madison.

### 2.1. Measures

#### 2.1.1. Coercion in Contraceptive Care Checklist

In our online survey, participants responded to open- and closed-ended questions about their health and contraceptive care. Based on theoretical conceptualizations of upward and downward contraceptive coercion [11,15] and our previous research interviewing stakeholders about regional family planning needs [19], we developed the five-item Coercion in Contraceptive Care Checklist. Figure 1 displays the items measuring the targeted aspects of contraceptive coercion dichotomously (0 = no, 1 = yes). Prior published research has used similar versions of these items [15]. We modified the phrasing of the items slightly based on content expert feedback and added an item based on recent research identifying providers’ hesitance to remove contraception at patients’ requests [10].

We assessed experiences of contraceptive coercion at two time points, first asking participants to “think back to the last time a healthcare provider talked to [them] about birth control” and then asking participants about experiences “at any healthcare visit in [their] lifetime”. We tabulated responses to the five items individually at both time points and also created dichotomous indicators of aggregate upward coercion, downward coercion, and any coercion. A few participants (4% across the upward coercion measures and 1% across the downward coercion measures) reported experiencing coercion at their last visit but answered “no” to the concordant measure about lifetime coercion. In these cases, we recoded participants who reported coercion at their last visit as having ever experienced that form of coercion.

Participants who reported experiencing coercion also answered follow-up questions about their experiences with contraceptive coercion. Specifically, we asked about the contraceptive method(s) that a provider had refused to give them, made them use, or made them keep using, providing a checklist of options and an “other” write-in option. We also provided an open-ended prompt inviting any participant who experienced coercion to “describe what happened in that healthcare encounter”.

#### 2.1.2. Quality of Family Planning and Discrimination in Healthcare

Participants also answered validated measures of the quality of their contraceptive care. Because theorists suggest that coercive practices often center providers’ concerns and public health initiatives over patient preferences [4,5], we included a measure of the quality of family planning care in our survey. Additionally, we included a measure of discrimination in healthcare because contraceptive coercion is often theorized to include discriminatory practices primarily targeting historically oppressed groups [10]. Because these scales measure similar but distinct aspects of contraceptive care, we used them in this study to establish construct validity for the Coercion in Contraceptive Care Checklist.

With the 11-item Interpersonal Quality of Family Planning Care (IQFP) Scale, we asked participants to think back to the last time a healthcare provider talked to them about birth control. Participants then rated specific qualities about that provider and the contraceptive counseling encounter, such as “taking my preferences about birth control seriously” and “letting me say what mattered to me about my birth control method”, on a 5-point Likert-type scale ranging from poor to excellent [20,21]. Responses to the 11 items were averaged, resulting in a score ranging from 1 to 5, where higher scores indicated better quality in contraceptive care. In this study sample, Cronbach’s alpha indicated that the IQFP items had high internal consistency (α = 0.971).

We used the 7-item Everyday Discrimination in Healthcare Scale to measure discrimination during contraceptive counseling. This measure was originally developed to measure chronic, routine discrimination in everyday life [22,23], and it has been adapted for healthcare settings [24,25,26,27]. For this measure, we asked participants to consider “all the times that [they had] talked to healthcare providers about birth control”. Participants reported how often a list of discriminatory experiences, such as “I had a provider act as if they thought I was not smart” and “I was treated with less respect than other people”, had happened to them on a 4-point Likert-type scale with response options of “never”, “once”, “2 or 3 times”, and “4 or more times”. We summed the 7 items, resulting in a score ranging from 0 to 21 where higher values indicated experiencing more discrimination in healthcare. In this study sample, Cronbach’s alpha indicated that the discrimination items had high internal consistency (α = 0.926).

### 2.2. Data Analysis

We began by tabulating the frequency of upward and downward contraceptive coercion in our sample (*N* = 1197). We also described the frequency of coercion related to specific contraceptive methods and used open-ended responses to demonstrate nuances in these experiences of contraceptive coercion.

Next, we conducted psychometric testing to assess the validity, reliability, and dimensionality of the Coercion in Contraceptive Care Checklist. We assessed the checklist’s reliability by calculating estimates of internal consistency using Cronbach’s alpha and inter-item correlations using the phi correlation coefficient. Then, we assessed the checklist’s construct validity by using *t*-tests to compare checklist responses to those on other measures of contraceptive care quality: average Interpersonal Quality of Family Planning Care Scale scores and summed scores on the Everyday Discrimination in Healthcare Scale. Finally, we used factor analysis to assess the checklist’s dimensionality, using robust weighted least-square estimation due to our dichotomous indicators.

## 3. Results

### 3.1. Sample Characteristics

Sample characteristics are provided in Table 1. The sample’s mean age was 32.78 years. Most of the sample identified their race/ethnicity as non-Hispanic white (65%), their sexual orientation as heterosexual (65%), and their gender identity as cisgender woman (95%). In line with our intentional oversampling of minority groups, other racial/ethnic groups, sexual orientations, and gender identities were represented in our sample at rates that are at or above rates in nationally representative samples [28,29]. Education level was evenly distributed, and the regional geographic distribution closely matched that of the United States overall (21% West U.S., 20% Midwest U.S., 19% Northeast U.S., and 40% South U.S.) [30].

### 3.2. Frequency of Contraceptive Coercion

Out of the 1197 participants who had ever talked to a healthcare provider about contraception, over 1 in 6 participants (*n* = 221, 18.46%) reported experiencing coercion during their last contraceptive counseling, and more than 1 in 3 (*n* = 506, 42.27%) reported experiencing contraceptive coercion at some point in their lifetime. At both the last visit and throughout participants’ lifetime, upward coercion (pressure to use birth control) was more common than downward coercion (pressure to not use birth control), with over a third of participants (34%) experiencing upward coercion at some point in their lives compared to one in six (16%) who reported experiencing downward coercion in their lifetime. The frequencies of these experiences are presented in Table 2.

Importantly, this analytic sample of 1197 people excludes those respondents who reported that they had never talked to a healthcare provider about contraception (n = 201). If we include those who never received contraceptive counseling in our estimates, the percentage of people who reported ever experiencing contraceptive coercion in their lifetime falls to 39% (31% upward coercion and 15% downward coercion). These frequencies can be used to compare estimates to other existing studies of contraceptive coercion that do not delineate between those who have and have not received contraceptive counseling.

In Figure 2, we show a breakdown of lifetime coercive experiences across 16 types of contraception, separated by upward and downward coercion. The birth control pill was the most common contraceptive method involved in both upward coercion and downward coercion. Overall, pressure to use or keep using the birth control pill was the most common manifestation of coercion perceived by U.S. patients across their lifetime (*n* = 175, 14.62% of the analytic sample and 42.89% of those reporting any upward coercion). Figure 2 highlights several other patterns in contraceptive coercion. For example, provider refusal of a patient’s desire for permanent contraception via tubal ligation, hysterectomy, or Essure (historically) was common (*n* = 41, 3.43% of the analytic sample and 21.35% of those reporting any downward coercion) compared to the refusal of other categories of contraception. Being refused desired birth control pills (*n* = 57, 4.76% of the analytic sample and 29.69% of those reporting any downward coercion), IUDs (*n* = 37, 3.09% of the analytic sample and 19.27% of those reporting any downward coercion), or implants (*n* = 17, 1.42% of the analytic sample and 8.85% of those reporting any downward coercion) were also relatively common.

### 3.3. Open-Ended Descriptions of Contraceptive Coercion

We used participants’ open-ended responses to add depth and nuance to their lived experiences of contraceptive coercion. Participants described some healthcare providers who did not listen to their desire not to use contraception. For example, one non-Hispanic white LGBT woman in her late 20 s said, “After giving birth to my daughter, I felt that I was forced into using birth control again. I said that I did not want to go back onto birth control because I didn’t like how it made my body felt [sic]. They didn’t really listen to me and sort of didn’t take no for an answer”. Patients reported that healthcare providers weighed the risk of pregnancy over patient concerns about symptoms. As one non-Hispanic Black heterosexual woman in her early 30 s reported,

I confided in my doctor that I don’t handle birth control hormones very well and the side effects were [too] day-altering. As expected, she dismissed my concerns and wanted to try a different formula of progesterone and estrogen or the copper IUD. I dismissed those options due to reasonable concerns and she told me that she expected to see me pregnant sooner than later if I didn’t pick a better option.


Some participants reported that they felt overwhelmed by the pressure to use a method of contraception. A Hispanic white heterosexual woman in her late 20 s stated, “I felt intimidated by the healthcare encounter I experienced with my healthcare [provider] because of the fact they were so adamant about me using birth control specifically the pills methods”. Some patients described experiencing pressure to start using contraception, even when seeking healthcare for other concerns. One non-Hispanic white heterosexual woman in her early 20 s stated, “I was 13 and only in to have them look at my sore throat and the doctor lady told me that I should get this shot for birth control even when I was not having or thinking about having sex. She made me uncomfortable”. Patients also reported that healthcare providers pushed them away from specific methods or brands of contraception and toward other methods or brands, often without providing reasoning or counseling patients on the differences between methods. One Hispanic heterosexual woman in her late 20 s reported, “I was interested in an IUD but she pushed me to get the hormonal one compared to the copper one without really giving me much info about the alternative”.

Open-ended responses related to downward contraceptive coercion included several instances of healthcare providers generally counseling patients not to use contraception. One non-Hispanic white LGBT nonbinary participant in their early 20 s made the following statement:

We had a difference in political opinion…when I talked to her about birth control options, so she was a little cold when talking to me about my options. I had to bring up the conversation and the patch because she only talked to me about the pill and IUD at first. Even then, she didn’t seem enthused to be discussing the topic with me and barely participated, even trying to sway me away from contraceptives entirely and just ‘use a condom’.


Patients also perceived that healthcare providers had pressured them not to use specific methods of contraception. Participants frequently described being denied permanent contraception via tubal ligation due to their age or based on assumptions that they may change their minds about having children in the future. Often, these assumptions were based on the idea that a future partner would want children. As one non-Hispanic white LGBT woman in her late 30 s reported, “I have had 2 successful healthy pregnancies at a very young age but also a tubal pregnancy a few years after and then a miscarriage in 2020. Despite being at an advanced age a tubal ligation was ‘out of the question’ because I ‘might get married again’. Which is ridiculous”. Patients were also refused LARC methods, as described by one non-Hispanic white heterosexual woman in her late 20 s: “I attempted to get an IUD after my second child but my hospital wouldn’t let me do it. I had to go to Planned Parenthood to get it instead”. Patients hoping to manage symptoms of menstruation were also pressured not to use contraception. A non-Hispanic Asian LGBT woman in her late 40s reported, “I brought up birth control to my doctor because I thought it might help reduce the severity of my periods/cramping. We had a discussion about it and she said that she would be wary of birth control and felt that I should just use Ibuprofen/Tylenol/other OTC medication”.

### 3.4. Measuring Contraceptive Coercion

Next, we conducted psychometric testing to assess the validity, reliability, and dimensionality of the Coercion in Contraceptive Care Checklist. To test the construct validity of the Coercion in Contraceptive Care Checklist, we assessed its relationship with validated measures of contraceptive care quality. As shown in Table 3, aggregate Coercion in Contraceptive Care Checklist indicators were significantly associated with participants’ average IQFP scores and their summed Everyday Discrimination in Healthcare scores. Specifically, participants who experienced downward contraceptive coercion reported lower average quality of their family planning care (*M* = 3.03, *SD* = 1.21) compared to those who did not experience downward coercion (*M* = 4.10, *SD* = 0.98; *t*[1194] = 7.54, *p* < 0.001). Those who experienced upward contraceptive coercion also reported lower average quality of their family planning care (*M* = 3.12, *SD* = 1.22) compared to those who did not experience upward coercion (*M* = 4.23, *SD* = 0.87; *t*[1194] = 14.76, *p* < 0.001). Relatedly, participants who experienced downward contraceptive coercion reported higher instances of discrimination in their contraceptive care (*M* = 7.11, *SD* = 5.54) compared to those who did not experience downward coercion (*M* = 2.27, *SD* = 3.77; *t*[1160] = −14.77, *p* < 0.001). Those who experienced upward coercion also reported higher instances of discrimination in their contraceptive care (*M* = 6.00, *SD* = 5.10) compared to those who did not experience upward coercion (*M* = 1.53, *SD* = 3.20; *t*[1160] = −18.27, *p* < 0.001).

Cronbach’s alpha demonstrated low internal consistency for both contraceptive coercion dimensions at both time points assessed (α = 0.23 for last visit downward coercion; α = 0.38 for last visit upward coercion; α = 0.35 for lifetime downward coercion; and α = 0.50 for lifetime upward coercion). Considering the limitations of Cronbach’s alpha, especially given the dichotomous response options and the very short format of the Coercion in Contraceptive Care Checklist [31], we also estimated reliability using the phi correlation coefficient. As shown in Table 4, the inter-item correlation between each of the items within a given domain of contraceptive coercion (i.e., upward coercion and downward coercion) was statistically significant (*p* < 0.001), although the phi coefficient values were relatively small (Φ = 0.13–0.33).

Finally, we conducted a confirmatory factor analysis to test the dimensionality of the lifetime Coercion in Contraceptive Care Checklist. The model fit indices indicated that the two-factor solution was a good fit to the data (*N* = 1197; Comparative Fit Index = 0.93; Tucker–Lewis Index = 0.81; root mean square residual = 0.08; standardized root mean square residual = 0.07) [32]. All five items exceeded our a priori criterion of loading at or above 0.40 (see Table 5). These results provide support for measuring provider-based contraceptive coercion across two dimensions: upward coercion and downward coercion.

## 4. Discussion

This study provides critical information about how U.S. patients experience contraceptive coercion and highlights opportunities for continued research and intervention to improve patient autonomy. We have focused our discussion on three key findings. First, we discuss and contextualize our finding that more than one in three patients have experienced coercion in their contraceptive care. Second, we appraise patterns and nuances in the types of coercion experienced, including the trends related to pressure to use and not use the birth control pill, IUDs, and permanent contraception. Third, we discuss our psychometric findings, highlighting possible applications of and improvements to the measurement of contraceptive coercion.

First, more than one in three patients in our sample perceived coercion in their contraceptive care at some point in their lifetime. This lifetime frequency among our national U.S. sample (39% of the total 1400 people sampled and 42% of those who have ever received contraceptive counseling) is almost identical to that reported in one of the only other existing quantitative studies of contraceptive coercion (37%) [15], even though that study sample was limited to the Appalachian region. Compared to the existing estimates in the Appalachian region, the current study also estimates very similar frequencies of upward (31% in our sample vs. 30% in the Appalachian region) and downward contraceptive coercion (15% in our sample vs. 16% in the Appalachian region). Although both studies are limited in their generalizability due to their purposive sampling strategies, they suggest that a non-trivial share of people experience some coercion in their contraceptive care, providing a starting point for the quantitative study of contraceptive coercion and highlighting the need for continued resources and research dedicated to understanding and eradicating coercion from contraceptive care.

Second, our findings highlight patterns and nuances in patients’ experiences of coercion in contraceptive care. For example, across their lifetimes, patients commonly perceived pressure to use the birth control pill as well as the refusal of tubal ligations. These findings are not surprising since the pill and permanent contraception are the two most common contraceptive methods in use [33], both of which require provider input via a prescription or medical procedure. Providers’ gendered and population-centered risk assessments may partially explain the frequency of upward coercion to use the birth control pill [34,35], underscoring the need for patient-centered, rather than population-centered, care that involves patient-provider information sharing and collaborative decision-making.

Additionally, the relative commonality of being refused a tubal ligation fits within the existing literature documenting providers’ hesitation and refusal to provide permanent contraception due to patient age and parity [9,12,13,14]. Patients also report system-level barriers to receiving permanent contraception, such as hospital guidelines for who is allowed to have a tubal ligation and when such procedures can and cannot take place [12]. Such hospital guidelines, along with state and federal regulations, are put into place to protect patients from receiving forced or coerced sterilization procedures [36], but this growing body of research suggests a need to reevaluate these regulations in order to protect patients from both upward and downward coercion related to permanent contraception.

Coercion related to IUDs was also relatively common compared to other methods, with both upward and downward coercion reported at similar rates. Patients’ perception of pressure to use IUDs is likely explained, at least in part, by efforts in the past few decades to increase LARC accessibility and public health enthusiasm about LARC promotion [3,4,5,34]. Additionally, participants stating that pressure to use LARC methods was “expected” speaks to the commonality of this issue and the norms and expectations around contraceptive counseling. In contrast, the finding that IUD refusal is one of the more common forms of perceived coercion indicates that other relational and/or structural factors may be at play. For example, providers can be more hesitant to provide IUDs for younger, non-monogamous, or nulliparous patients, reflecting outdated practice guidelines for IUD eligibility [37,38].

Providers’ racial biases, both explicit and implicit [39], as well as structural racism, which causes inequities in who has physical and financial access to contraception [40], also likely play a role in continued stratified reproduction and explain how and why coercion appears in both upward and downward directions. Additionally, patients may perceive structural issues such as a provider’s inability to place an IUD at the same appointment as a coercive refusal of care. These nuances and patterns highlight opportunities for more patient-centered care, although continued research is needed to document the causes and impacts of these experiences.

Third, this study has implications for the continued measurement of contraceptive coercion. On our five-item Coercion in Contraceptive Care Checklist, people who experienced contraceptive coercion were more likely to report that they had experienced a lower quality of family planning care and a higher frequency of discrimination in their contraceptive care. This provides evidence of the measure’s construct validity. Statistically significant inter-item correlations also provide evidence of the measure’s reliability although relatively low estimates of Cronbach’s alpha suggest that the items may lack internal consistency reliability and/or that the measure contains too few items to adequately obtain a high alpha estimate. Factor analysis supports the theoretical model of coercion in contraceptive care as a two-factor construct with dimensions of upward and downward coercion.

Taken together, these results provide evidence of the Coercion in Contraceptive Care Checklist’s validity, reliability, and dimensionality while also suggesting avenues for future testing and refinement [41]. Other researchers can use and build upon our five-item Coercion in Contraceptive Care Checklist. In addition to being used in health and social science surveys to understand the prevalence and impact of contraceptive coercion, this measure could be used for quality improvement in clinical settings.

The measure could also benefit from additional testing followed by refinement, as needed. In particular, cognitive interviewing could help further evaluate the measure’s validity and reliability and suggest possible improvements to the items. For example, it is not known how patients may respond differently to items using more direct phrasing such as “pressure”, “force”, or “coercion” rather than our more general language (e.g., “Made me feel…”). It is possible but not known whether varying this phrasing could capture a range of coercive experiences from more subtle to more overt. Additionally, cognitive interviewing could help establish whether patients are considering structural as well as interpersonal aspects of contraceptive care in their interpretation of the items in the Coercion in Contraceptive Care Checklist. For example, some participants may endorse the items assessing downward contraceptive coercion when they were in fact denied contraception for a medically valid reason or due to an access issue unrelated to coercion. Cognitive interviewing could help investigate such nuances in participants’ responses. Finally, additional tests, such as test-retest reliability, could also help us better understand the reliability of the measure. Adding additional items to the checklist could improve estimates of internal consistency reliability, although this is not warranted unless cognitive interviewing or other qualitative work identifies additional theoretical manifestations of coercion in contraceptive care.

### Limitations and Strengths

This study has several limitations. First, we used a non-probability sampling method by recruiting participants through Prolific. Despite the benefits of this approach for study cost and completion time, this sampling method limits the representativeness and generalizability of the study findings. Second, as there is no current gold standard for measuring contraceptive coercion, establishing criterion validity is hindered by the lack of a comparison scale. Above, we have highlighted several opportunities for continued research that could help to further validate the Coercion in Contraceptive Care Checklist.

This study also has important strengths. This is the first study that investigates contraceptive coercion quantitatively in a national sample, providing important information about the validity of a novel survey instrument and evidence of the frequency and manifestations of contraceptive coercion. We also intentionally recruited a diverse sample across race, ethnicity, sexual orientation, and gender identity. This allows us to build evidence about contraceptive coercion for these populations that are both difficult to reach in survey research and conceptualized as at increased risk of contraceptive coercion. This also has important implications for measurement development, as measures are commonly developed using samples that lack diversity and therefore may not be appropriate for use with non-white, non-heterosexual participants.

## 5. Conclusions

These findings provide critical information about contraceptive coercion, including its frequency in a national sample. We also describe nuances in patient experiences of contraceptive coercion and provide evidence of the validity, reliability, and dimensionality of the Coercion in Contraceptive Care Checklist. Finally, we suggest avenues for future testing and refinement and highlight opportunities for continued research and intervention to improve patient autonomy.

## Figures and Tables

**Figure 1 ijerph-21-00750-f001:**
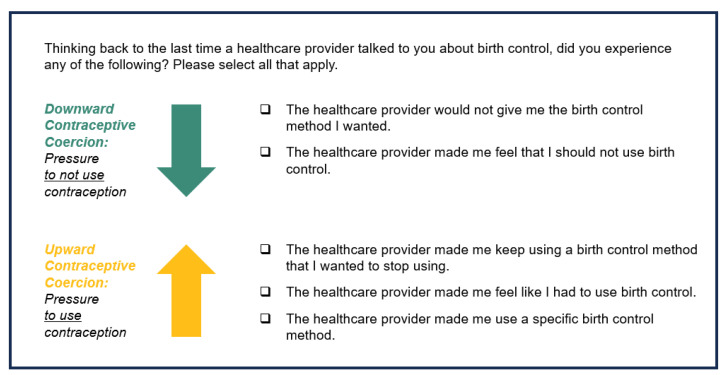
The Coercion in Contraceptive Care Checklist, shown with theoretical descriptions. Figure Note: The text above assesses coercion at the most recent contraceptive counseling. When assessing lifetime coercion in contraceptive care, the text reads “At any healthcare visit in your lifetime, have you ever experienced any of the following? Please select all that apply”. Additionally, options read “A healthcare provider” instead of “The healthcare provider”.

**Figure 2 ijerph-21-00750-f002:**
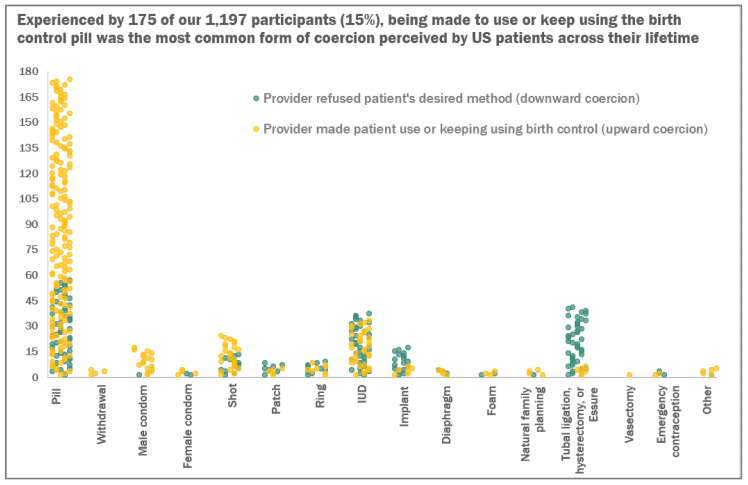
Methods of contraception patients reported being coerced to use or not use in a 2023 sample of U.S. reproductive-aged people assigned female at birth (*N* = 1197).

**Table 1 ijerph-21-00750-t001:** Characteristics of this 2023 sample of U.S. reproductive-aged people assigned female at birth who had ever talked to a healthcare provider about birth control (*N* = 1197).

Variable	*n* (%)	*M* (*SD*)
Age (18–49 years)		32.78 (8.08)
Race/ethnicity		
Hispanic	101 (8.44%)	
Non-Hispanic Asian	83 (6.93%)	
Non-Hispanic Black	182 (15.20%)	
Non-Hispanic white	783 (65.41%)	
Non-Hispanic mixed race or other	48 (4.01%)	
Gender identity		
Cisgender woman	1137 (94.99%)	
Trans man AFAB	16 (1.34%)	
Nonbinary AFAB	44 (3.68%)	
Sexual orientation		
Heterosexual	779 (65.08%)	
Gay/lesbian	53 (4.43%)	
Bisexual	245 (20.47%)	
Asexual, pansexual, queer, questioning, or other	120 (10.03%)	
Education level		
High school or less	141 (11.78%)	
Associate degree or some college	390 (32.58%)	
Bachelor’s degree	444 (37.09%)	
Graduate school	222 (18.55%)	
Time since the last contraceptive counseling		
In the past year	559 (46.70%)	
1–3 years	412 (34.42%)	
4–6 years	136 (11.36%)	
7+ years	90 (7.52%)	

**Table 2 ijerph-21-00750-t002:** Coercion in contraceptive care in a 2023 sample of U.S. reproductive-aged people assigned female at birth who had ever talked to a healthcare provider about birth control (*N* = 1197).

Coercion in Contraceptive Care	At Last Counseling	Ever
Any coercion		
No	976 (81.54%)	691 (57.73%)
Yes	221 (18.46%)	506 (42.27%)
Downward coercion		
No	1146 (95.74%)	1005 (83.96%)
Yes	51 (4.26%)	192 (16.04%)
Would not give me the birth control method I wanted	33 (2.76%)	144 (12.03%)
Made me feel that I should not use birth control	22 (1.84%)	79 (6.60%)
Upward coercion		
No	1014 (84.71%)	789 (65.91%)
Yes	183 (15.29%)	408 (34.09%)
Made me keep using a birth control method that I wanted to stop using	27 (2.26%)	111 (9.27%)
Made me feel like I had to use birth control	127 (10.61%)	275 (22.97%)
Made me use a specific birth control method	69 (5.76%)	190 (15.87%)

Note: The analytic sample (*N* = 1197) included only those participants who had ever talked to a healthcare provider about birth control. Rates of contraceptive coercion among the full sample (*N* = 1400) are provided in the text.

**Table 3 ijerph-21-00750-t003:** Interpersonal Quality of Family Planning Care (IQFP) and Everyday Discrimination in Healthcare, in a 2023 Sample of U.S. reproductive-aged people assigned female at birth who had ever talked to a healthcare provider about birth control (*N* = 1197).

Coercion in Contraceptive Care Checklist ^1^	*M* (*SD*)	
	Average IQFP Score ^2^ (*n* = 1196)	Sum of Everyday Discrimination in Healthcare ^3^ (*n* = 1162)
Any coercion		
No	4.27 (0.84) *	1.06 (2.52) *
Yes	3.15 (1.21) *	5.77 (5.10) *
Downward coercion		
No	4.10 (0.98) *	2.27 (3.77) *
Yes	3.03 (1.21) *	7.10 (5.54) *
Upward coercion		
No	4.23 (0.87) *	1.53 (3.20) *
Yes	3.12 (1.22) *	6.00 (5.10) *

* *p* < 0.001 in independent-sample *t*-tests. ^1^ Contraceptive coercion was assessed at the last counseling for comparisons with the IQFP and at lifetime for comparisons with Everyday Discrimination in Healthcare in order to match the recommended timeframes for these established measures. ^2^ The IQFP uses 11 items to measure the quality of family planning care at the last contraceptive care visit. Average scores ranged from 1 to 5 (*M* = 4.06), where higher scores indicate better quality care. ^3^ The Everyday Discrimination in Healthcare measure uses 7 items to measure discrimination ever experienced in contraceptive care. Sum scores ranged from 0 to 21 (*M* = 3.04), where higher scores indicate experiencing more discrimination.

**Table 4 ijerph-21-00750-t004:** Inter-item correlations among items in the Coercion in Contraceptive Care Checklist, in a 2023 sample of U.S. reproductive-aged people assigned female at birth who had ever talked to a healthcare provider about birth control (*N* = 1197).

Coercion in Contraceptive Care Checklist, at Last Counseling	Phi Correlation Coefficient				
	dcc1	dcc2	ucc1	ucc2	ucc3
Downward coercion					
dcc1: Would not give me the birth control method I wanted	-	0.129 *	-	-	-
dcc2: Made me feel that I should not use birth control	0.129 *	-	-	-	-
Upward coercion					
ucc1: Made me keep using a birth control method that I wanted to stop using	-	-	-	0.130 *	0.156 *
ucc2: Made me feel like I had to use birth control	-	-	0.130 *	-	0.241 *
ucc3: Made me use a specific birth control method	-	-	0.156 *	0.241 *	-
Coercion in Contraceptive Care Checklist, lifetime	Phi Correlation Coefficient				
	dcc1	dcc2	ucc1	ucc2	ucc3
Downward coercion					
dcc1: Would not give me the birth control method I wanted	-	0.222 *	-	-	-
dcc2: Made me feel that I should not use birth control	0.222 *	-	-	-	-
Upward coercion					
ucc1: Made me keep using a birth control method that I wanted to stop using	-	-	-	0.229 *	0.326 *
ucc2: Made me feel like I had to use birth control	-	-	0.229 *	-	0.241 *
ucc3: Made me use a specific birth control method	-	-	0.326 *	0.241 *	-

* *p* < 0.001.

**Table 5 ijerph-21-00750-t005:** Confirmatory factor analysis of the Coercion in Contraceptive Care Checklist in a 2023 sample of U.S. reproductive-aged people assigned female at birth who had ever talked to a healthcare provider about birth control (*N* = 1197).

Coercion in Contraceptive Care Checklist Items	Factor Loadings	R^2^
Downward Coercion		
Would not give me the birth control method I wanted	0.972 **	0.944 **
Made me feel that I should not use birth control	0.492 **	0.242 *
Upward Coercion		
Made me keep using a birth control method that I wanted to stop using	0.813 **	0.662 **
Made me feel like I had to use birth control	0.474 **	0.224 **
Made me use a specific birth control method	0.808 **	0.653 **

* *p* < 0.01; ** *p* < 0.001.

## Data Availability

Data are not publicly available due to privacy and ethical restrictions.

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
