# Peer review of "Healthcare Provider-Based Contraceptive Coercion: Understanding U.S. Patient Experiences and Describing Implications for Measurement"

_ijerph, 2024, doi:10.3390/ijerph21060750_

Round 1
Reviewer 1 Report
Comments and Suggestions for Authors
Thank you for the opportunity to review “Healthcare Provider-Based Contraceptive Coercion: Understanding U.S. Patient Experiences and Describing Implications for Measurement”, which is analyzes cross-sectional survey data on contraceptive coercion, and conducts factor analysis on a new measure of this concept.
This is such important work, and a very rigorous study and well-written manuscript. I look forward to citing it in my own work.
I am tripping over the conceptualization of “The healthcare provider would not give me the birth control method I wanted” as downward coercion – I can envision someone answering yes to this when they say they want to use condoms or a diaphragm or even pills, and the provider pushes them to use a LARC, which would be upward coercion. Or many times I am unable to provide someone the method they want due to medical contraindications – patients definitely perceive this as “she wouldn’t give me the method I wanted” (this is how they describe it to me), but it does not fit the conceptualization of coercion. Perhaps some of this nuance can be added to the Limitations section.
Did authors use any other methods of ensuring data quality from a crowd-sourced sample, like attention checks? Spot checking surveys for pattern or non-sensical responses?
For the IQFP and the Everyday Discrimination Scale, a few examples of the “specific qualities” and “discriminatory experiences” would be helpful.
Section 2.2 changes verb tense a bit. I think it should be past tense?
I’m curious about the intentional oversampling of some groups – this is an excellent idea but why not sample even more equitable numbers such as 1/5 Asian, 1/5 Black, 1/5 Hispanic etc…
Figure 2 is a really nice way to display this data. What I am left wondering, however, is who was refused and who was coerced into the various methods, especially pills and sterilization? Were there any racial/ethnic or other correlates of these experiences? If this data was too small for this kind of analysis, perhaps suggest it for future work.
I also was left wondering what is the significance of being “made to continue” a method like pills or condoms, which the patient has full power to stop using, as opposed to refusal to remove an IUD or implant.
The data on being made to use withdrawal method is questionable – though many patients use it and some use it effectively, I cannot imagine any provider encouraging any patient to use this method. Having said that, some of the qualitative responses clearly indicate some providers are opposed to hormonal and other “modern” methods of birth control, so perhaps these same providers are encouraging withdrawal? If not already done, I would encourage a close review of these surveys to make sure other responses seem reasonable/reliable.
The free-text responses are really helpful in getting a sense for the coercion, barriers, and generally bad care people experience.
2nd paragraph in Discussion – I had to go back to Table 2 a few times to figure out why the numbers did not match – the “39% of the total 1,400 people sampled” is helpful, but maybe this can be emphasized more in this paragraph so the reader doesn’t forget that you calculated both ways (full sample and those who had counseling)? Or maybe include the full sample numbers in Table 2, as well?
P11, Line 410-411: To add context to the finding about reluctance to provide IUDs – this also may reflect outdated recommendations to not give IUDs to nulliparous women. Some providers are still stuck in those old recommendations.
Line 415-416: yes! Inability to provide the IUD at the same appointment may be a billing issue, may be a logistical/scheduling issue, and also sometimes a Title X clinic may not have access to all options of birth control – for many years I couldn’t offer hormonal IUDs to patients because our clinic couldn’t stock them due to some kind of regulatory issue. Now I can offer one kind of hormonal IUD, but not others.
Line 459: Change “We are” to “This is” or “We have conducted”.
Great study! Great manuscript!
Reviewer 2 Report
Comments and Suggestions for Authors
Abstract – Would add in line 11 "established quantitative method". There is qualitative studies describing coercion but not as much in the quantitative space.
Introduction – Great laying out of coercion landscape of research and glad upward/downward framing was used.
Methods - No comments.
Results – Wondering if the language should change "tubal ligation" to "permanent contraception" in alignment with ACOG/SFP language.
Originality / Contribution – This will be a major contribution for the area of contraceptive coercion and was thoughtfully brought about via mixed methods to test a new scale.
